# Pathophysiology of Status Epilepticus Revisited

**DOI:** 10.3390/ijms26157502

**Published:** 2025-08-03

**Authors:** Rawiah S. Alshehri, Moafaq S. Alrawaili, Basma M. H. Zawawi, Majed Alzahrany, Alaa H. Habib

**Affiliations:** 1Department of Clinical Physiology, Faculty of Medicine, King Abdulaziz University, Jeddah 22252, Saudi Arabia; bzawawi@kau.edu.sa (B.M.H.Z.); ahabib@kau.edu.sa (A.H.H.); 2Department of Neurology, Faculty of Medicine, King Abdulaziz University, Jeddah 22252, Saudi Arabia; alruily@kau.edu.sa (M.S.A.);

**Keywords:** status epilepticus, pathophysiology, pharmacoresistance, epileptogenesis

## Abstract

Status epilepticus occurs when a seizure lasts more than five minutes or when multiple seizures occur with incomplete return to baseline. SE induces a myriad of pathological changes involving synaptic and extra-synaptic factors. The transition from a self-limiting seizure to a self-sustaining one is established by maladaptive receptor trafficking, whereby GABA_A_ receptors are progressively endocytosed while glutamatergic receptors (NMDA and AMPA) are transported to the synaptic membrane, causing excitotoxicity and alteration in glutamate-dependent downstream signaling. The subsequent influx of Ca^2+^ exposes neurons to increased levels of [Ca^2+^]i, which overwhelms mitochondrial buffering, resulting in irreversible mitochondrial membrane depolarization and mitochondrial injury. Oxidative stress resulting from mitochondrial leakage and increased production of reactive oxygen species activates the inflammasome and induces a damage-associated molecular pattern. Neuroinflammation perpetuates oxidative stress and exacerbates mitochondrial injury, thereby jeopardizing mitochondrial energy supply in a state of accelerated ATP consumption. Additionally, Ca^2+^ overload can directly damage neurons by activating enzymes involved in the breakdown of proteins, phospholipids, and nucleic acids. The cumulative effect of these effector pathways is neuronal injury and neuronal death. Surviving neurons undergo long-term alterations that serve as a substrate for epileptogenesis. This review highlights the multifaceted mechanisms underlying SE self-sustainability, pharmacoresistance, and subsequent epileptogenesis.

## 1. Introduction

A seizure is a paroxysmal hyperexcitable hypersynchronous neuronal discharge originating in the gray matter of the cerebral cortex and the subcortex [1]. Seizures are usually transient and self-terminating discharges that may be confined to a focal area or may spread to affect both hemispheres [2,3]. Brief self-terminating seizures have not been proven to cause neuronal injury [4]. However, if endogenous mechanisms fail to terminate a seizure within minutes, a series of maladaptive changes ensue [2]. When a seizure is abnormally prolonged or is repeated without a complete return to baseline, it is called status epilepticus (SE) [1,2,3]. SE is a common neurological emergency with an estimated incidence of 10–40 cases per 100,000 population [2,3]. It is associated with time-dependent alterations resulting in significant morbidity and mortality. The mortality rate associated with SE is reportedly between 20% and 30%, with radiographic and biochemical evidence of neuronal death following refractory SE [2,5]. This evidence is more abundant for the most common semiology of SE, convulsive SE (CSE) [2,5,6]. It is estimated that 10% of patients who survive CSE end up with variable functional and cognitive disabilities [7]. Still, prospective studies highlight cumulative delays and underdosing in the management of SE despite protocols recommending timely administration of gamma-aminobutyric acid (GABA)ergic benzodiazepine (BZD) therapy and early escalation to non-BZD therapy [2,3]. Despite the increased availability and variety of antiseizure medications (ASMs), the outcome of SE has not improved but has, in fact, worsened in recent years [8]. Understanding the underlying intricate molecular mechanisms of incessant seizure sustainability and pharmacoresistance might have significant implications for the development of new therapeutics and treatment guidelines. This article explores the mechanisms underlying the sustainability and pharmacoresistance of seizure activity in SE, as well as the subsequent long-term alterations contributing to epileptogenesis. Systemic complications of SE, such as acidosis and changes in blood pressure [9,10,11], are beyond the scope of this review.

## 2. Definition and Classification of Seizure Activity

A single seizure is a paroxysmal, short-lived alteration of neurological function, with significant inter-seizure intervals [1,12]. Often, seizures may recur serially with shorter inter-seizure intervals in what is termed seizure clusters or acute repetitive seizures. There is no consensus as to what marks a significantly short inter-seizure interval for repetitive seizures to be considered separate single seizures or seizure clusters [4,13]. However, research into the temporal distribution of seizures suggests that an inter-seizure interval of fewer than eight hours reflects seizures originating from a single ictal focus and are thus considered clusters [14]. Seizure clusters differ from recurrent seizures of SE in that there is a recovery between seizures. Still, seizure clusters represent a possible latent ictal focus that has the potential to evolve into SE due to synaptic potentiation from an initial seizure [4,14,15]. The likelihood of seizure clusters progressing into SE may be predicted based on the length of seizures and clinical history [14]. SE had historically been defined as a seizure lasting more than thirty minutes [16]. However, this definition was deemed impractical since experimental evidence has delineated two different time points, which differ according to seizure type: a time point that marks abnormal prolongation of seizure activity, and a time point that signifies a risk of neurological injury. Hence, the International League Against Epilepsy (ILAE) proposed a conceptual definition for SE with two operational dimensions: the first time point (t1) marks the time after which seizure activity is abnormally prolonged and is unlikely to self-terminate while the second time point (t2) marks the time needed to sustain neuronal injury and long-term consequences [1,2]. Based on experimental and clinical evidence, t1 has been defined as five minutes for tonic–clonic seizures, ten minutes for focal seizures, and ten to fifteen minutes for absence seizures, while t2 has been defined as thirty minutes for tonic–clonic SE, sixty minutes for focal SE, but has not been determined for absence SE [2]. Additionally, there exists an operational timeframe that differentiates SE into four stages: impending, established, refractory, and super-refractory [17]. Impending and established SE are defined as continuous or intermittent seizures without returning to baseline lasting more than 10 min and 30 min, respectively [17]. Nearly a quarter of SE cases become refractory to appropriate first- and second-line ASMs and require induction of anesthesia [18]. Nearly half of these refractory seizures persist for more than 24 h after induction of anesthesia, at which point SE is super-refractory [18].

## 3. Models of Seizure Activity

Much of our understanding of the pathophysiology of SE is based on cellular and translational disease models of epileptogenesis, summarized in Figure 1. Although postmortem histopathological analysis of human tissue has provided insights into the long-term sequelae of SE, more substantial evidence is inferred from studies utilizing in vivo models that often parallel generalized CSE (GCSE) [19]. Therefore, it is important to revisit the fundamentals of SE modeling. Induction of in vivo seizures can be achieved chemically, by electrical stimulation, or by genetically induced mutations [19,20]. SE can be induced using chemoconvulsants that either enhance glutamatergic or cholinergic neurotransmission or inhibit GABAergic neurotransmission. The most commonly used chemoconvulsants are kainic acid, pilocarpine, and lithium-pilocarpine [11,20]. Kainic acid (KA) is an agonist for the KA and the α-amino-3-hydroxy-5-methyl-4-isoxazolepropionic acid (AMPA) subtypes of ionotropic glutamate receptors that can be administered via intracerebroventricular, intrahippocampal, or intraperitoneal injections to induce limbic seizures in rats and mice [21]. Pilocarpine is a muscarinic cholinergic agonist that can be administered via intraamygdaloid, intrahippocampal, or intraperitoneal injections to induce limbic seizures in rats and mice. Cholinergic hyperstimulation enhances glutamatergic excitation, and it is, therefore, likely that the long-term sequelae seen in this model are secondary to glutamate release [22]. To improve the success rate of pilocarpine induction while lowering the mortality associated with high doses of pilocarpine, lithium chloride may be administered in rats hours before administration of pilocarpine in the lithium-pilocarpine model [11]. In KA, pilocarpine, and lithium-pilocarpine models, the epileptiform discharges originate in the limbic system and propagate to other brain areas, resulting in GCSE with long-term sequelae reminiscent of those seen in human CSE [21]. However, the pilocarpine and lithium-pilocarpine models are often preferred over the KA models for pharmacological experiments as the variable animal strains are more homogenously sensitive to ASMs, and they exhibit seizures that become progressively pharmacoresistant if left untreated [11]. It is noteworthy, however, that pilocarpine induces greater neocortical damage than KA [11]. Less commonly used chemoconvulsants include cholinergic agents such as organophosphate and flurothyl, GABA antagonists such as bicuculline, picrotoxin, and pentylenetetrazole, and modulators of *N*-methyl-D-aspartate (NMDA) receptors such as d,l-homocysteine thiolactone, glutamate, and glycine [11,23]. Chemical induction has a direct excitotoxic effect, thereby obscuring the line between direct neuronal damage and seizure-induced neuronal damage, and carries a relatively high risk of respiratory arrest, especially in mature rodents [20]. Electrical stimulation has the advantage of sparing the animal from direct excitotoxicity, although electrode implantation can be complex and time-consuming [11,24]. The electrical stimulation models of SE are rat models with electrodes implanted to stimulate the afferent pathway to the hippocampus in the perforant pathway stimulation (PPS) model or direct continuous stimulation of the hippocampus in the self-sustaining limbic (SSL) stimulation model [11]. Overall, electrical induction is associated with lower mortality and a longer duration to induce epileptogenesis than chemical induction [23]. Genetic models of epilepsy, including phosphatase and tensin homolog (*PTEN*) knockout mice and tuberous sclerosis complex1 (*TSC1*) knockout mice, are often used to study chronic epilepsy and epilepsy syndromes rather than SE [25,26]. In vitro models, on the other hand, are cost-effective and time-effective models with the ability to control many of the variables and restrict the effect of confounding factors without the ethical dilemmas associated with in vivo modeling [19,27]. To induce epileptiform discharges in neurons maintained in primary culture, or in an isolated brain slice from healthy or epileptic tissue, a medium containing a proconvulsant is applied to the culture or slice, such as the K^+^ channel blocker 4-Aminopyridine (4-AP) or low Mg^2+^ media. In vitro electrophysiological recordings can be made using glass microelectrodes according to detailed protocols [19,28,29]. Still, in vitro modeling is not without limitations, as isolation from other parts of the brain blocks the input and feedback from circulating and surrounding factors, thereby affecting its translation potential to humans [11,27].

Although nearly 25% of SE are nonconvulsive, models of nonconvulsive SE (NCSE) are lacking [23]. SE is a heterogeneous disorder that encompasses many causes and seizure phenotypes; a translational gap is inevitable, as it is unlikely that a single model can capture the full spectrum of SE characteristics. A smaller dose of pilocarpine has been proposed to induce NCSE in lithium chloride-pretreated rats [30]. However, the use of a lower dose in this model may overlook the heterogeneous nature of NCSE, which may be, in some cases, an advanced stage of CSE [31].

## 4. Pathophysiology of Status Epilepticus

SE is not a syndrome in and of itself but rather a state that arises from an underlying pathology. The etiology of SE is variable and is considered the main determinant of outcome [32]. The pathophysiology of SE is complex, involving synaptic and extrasynaptic mechanisms underlying seizure self-sustainability, pharmacoresistance, and neuronal injury, detailed below.

### 4.1. Aberrant Neurotransmission

Seizures are epileptiform discharges from the gray matter of the cerebral cortex or the subcortex caused by an imbalance between neuronal excitation and inhibition, leading to excessive firing of neurons and heightened reactivity to stimulation, termed hyperexcitability and simultaneous firing of numerous neurons at the same rate, termed hypersynchrony, modulated by voltage-gated ion channels in neuronal cell bodies, axons, and dendrites as well as gap junctions [12,33]. While it is widely accepted that seizure activity, including SE, is hypersynchronous discharge, clinical synchronization dynamics of EEG, magnetoencephalography, and intracellular recordings reveal heterogeneous synchronization throughout the course of SE [34]. Some researchers have documented desynchronization at the height of SE activity and have argued that recovery from SE is marked by a transition from a desynchronous phase to a hypersynchronous phase in which neurons fire simultaneously [35,36]. Transition to a hypersynchronous phase is regarded by some as a “prerequisite” for the termination of seizure activity. The desynchronization hypothesis is detailed in a previous review [36].

At the cellular level, the influx of Ca^2+^ causes the opening of voltage-gated Na^+^ channels, the subsequent influx of Na^+^, and the generation of recurring action potentials. This is normally followed by a hyperpolarizing afterpotential, which is mediated by GABAergic inhibition generated by interneurons and increased Na^+^/K^+^ ATPase activity aimed at restoring ionic balance [37]. Seizure propagation from an epileptic focus to surrounding areas is normally inhibited by GABAergic hyperpolarization due to Cl^−^ influx through ligand-gated channels of surrounding inhibitory neurons [12,37,38]. The ictal discharge preceding seizure termination is characterized by large-amplitude bursts followed by large inter-burst intervals. It is hypothesized that the progressive increase in the amplitude and duration of synchronous post-burst depression dampens the potential for burst re-excitation [39]. A model of pyramidal cells and fast-spiking inhibitory interneurons was studied using a conductance-based approach to explore the ionic mechanisms of spontaneous seizure termination, revealing that a progressive increase in [Na^+^]i mediates spontaneous seizure termination [40]. However, the findings of this study have not been substantiated by further research. Other mechanisms have been linked to seizure termination, such as K+ currents activated by Ca^2+^ and Na^+^ entry and associated alterations in the transmembrane gradients of ions, especially of K^+^, leading to blockage of depolarization [38,39,41]. It is likely that seizure termination is the result of a synergy of the aforementioned mechanisms [37]. When these mechanisms fail to terminate ictal discharge, a series of alterations ensue, leading to seizure self-sustainability [35]. Failure of the mechanisms of seizure termination is likely linked to the intensity and duration of the triggering stimulus. Contrary to single seizures, the majority of SE episodes are provoked and not solely attributed to epilepsy [1,41,42]. Additionally, low serum concentration of ASMs in patients with epilepsy as an etiology of SE is associated with a better outcome than other etiologies [17]. In these cases, surround inhibition may be overcome by enhanced neurotransmitter release due to excessive presynaptic accumulation of Ca^2+^ in presynaptic terminals and excessive elevation of [K^+^]o, which decreases the threshold of neuronal excitation due to accumulation of K^+^ and depolarization of the membrane potential. Additionally, accumulation of [K^+^]o reduces the electrochemical driving force for the K^+^-Cl^−^ co-transporter (KCC2), thereby reducing chloride extrusion and inhibitory GABAergic signaling [41,43]. Accumulation of [K^+^]o is further exacerbated by reduced astrocytic inwardly rectifying potassium (Kir) channels containing Kir4.1 subunits Kir4.1 channels which normally buffer [K^+^]o and maintain glutamate homeostasis [38,39]. Therefore, seizure sustainability and subsequent increase in [K^+^]o creates a positive feedback loop to make the network more excitable. Additionally, depolarization induces the activation of the NMDA receptor, which causes further Ca^2+^ influx, creating a positive feedback loop [38]. Failure of surround inhibition leads to depolarization of neighboring neurons, which spreads contiguously and across cortical and subcortical regions via cortico-cortical synaptic connections and association pathways [12,37]. Propagation of seizure to cortical, thalamic, and brain stem centers is responsible for the loss of consciousness associated with generalized seizures [12]. It is likely that other mechanisms are involved in the synchronization and propagation of seizure activity, such as electrical synaptic communication, ephaptic interactions, and activity-dependent ionic changes [43]. Additionally, elevated [K^+^]o and reduced astrocytic K^+^ buffering lead to astrocytic depolarization and reversed electrogenic glutamate uptake, facilitating seizure transition into the maintenance phase [44]. The maintenance phase of SE is characterized by synaptic potentiation, caused by internalization and downregulation of the inhibitory GABA_A_ receptors and runaway excitation due to overexpression of excitatory receptors (AMPA and NMDA receptors) [45]. Electrical and chemical stimulation models show internalization of synaptic BZD-sensitive GABA_A_ (containing *β* 2/3 and/or *γ* 2 subunits) receptors into endosomes concurrent with mobilization of the GluR1 subunits of NMDA receptors to the synaptic membrane, resulting in a decrease in the number of GABA_A_ receptors and an increase in the number of NMDA receptors per somatic granule cell synapse. In addition, this is associated with NMDA-dependent mobilization of BZD-insensitive GABA_A_ receptors (containing *α* 4 subunits) and at least three subunits of AMPA (GluR1, GluR2, and GluR3) to the synaptic membrane [46,47,48]. These AMPA subtypes mediate the initial, Na^+^-dependent, rapid depolarization of postsynaptic neurons, which detaches Mg^2+^ from the NMDA pore, allowing for further activation of NMDA receptors [47]. This potentiating effect of neurotransmitter receptor trafficking makes seizure activity self-sustaining, independent of the triggering stimulus, as seen in electrical stimulation models where seizures persist after cessation of stimulation [46,49]. Receptor trafficking takes place within 10–30 min of CSE onset, causing time-dependent resistance to ASMs that modulate GABA activity, such as BZD [9,45]. Lithium-pilocarpine models show sensitivity to BZD when administered within 10 min of seizure onset, but the sensitivity of GABA receptor currents decreases progressively over time [50]. Some brain areas may be more prone to earlier receptor trafficking than others. The hippocampus is thought to be subject to earlier internalization of GABA_A_ receptors than the amygdala, piriform cortex, and endopiriform nucleus [51]. Seizure activity is further facilitated by failure of GABA synthesis in the substantia nigra, which may be caused by increased consumption of the precursor glutamate or dysfunction of glutamate transporters induced by the excess release of glutamate [52,53] and loss of somatostatin-containing interneurons and mossy cells that normally activate inhibitory neurons, as evidenced in PPS models [24]. Autophagy normally governs neurotransmitter release through the disintegration of synaptic vesicles and autophagy-dependent synaptic clustering of GABA_A_ receptors. However, SE is associated with hyperactivation of the mammalian target of rapamycin (mTOR), likely due to disturbances in the ubiquitin-proteasome system, leading to suppression of autophagy. Autophagy failure results in intracellular clustering of GABA_A_ receptors whilst blocking the degradation of glutamate receptors, resulting in an increased ratio of excitatory synaptic receptors to inhibitory receptors [38]. Prolonged seizure activity not only affects the distribution of receptors but might also affect their expression; in vitro hippocampal models show activity-dependent alteration of AMPA receptor expression [29]. The maintenance phase is consolidated by maladaptive changes in protein kinases and neuropeptides [46]. Hippocampal cultures from transgenic models and chemically induced models show downregulation of the inhibitory peptides dynorphin, galanin, somatostatin, and neuropeptide Y, and upregulation of the ictogenic tachykinins such as substance P and neurokinin B [17,54,55]. Differential expression of several other proteins is described in SE models, including synaptic plasticity-related proteins, further tilting the balance toward ictogenesis [56,57,58].

### 4.2. Neuroinflammation

Neuroinflammation is at the center of SE pathogenesis as both a consequence and a facilitator of neuronal injury, beyond the infectious and immune-mediated etiologies [59,60,61]. Brain tissues from SE models show increased expression of the astroglial biomarkers S100B and glial fibrillary acidic protein (GFAP) and inflammation biomarkers such as IL-1ß, IL-6, and High Mobility Group Box 1 (HMGB1), as early as two hours after SE induction [61,62,63,64,65,66]. Clinically, the levels of these biomarkers have been found to correlate with SE outcome [65,67,68]. Experimentally, neuronal injury seems to correlate more with astrogliosis and cytokine expression than the severity of SE, especially in immature rodents [69].

Prolonged excessive neuronal activity and concurrent altered metabolism promote a stress response and subsequent morphological and molecular alterations of astroglia into reactive and toxic phenotypes known as A1 astrocytes and M1 microglia. Neuronal injury induces the release of pro-inflammatory molecules by damaged neurons, such as HMGB1 and damage-associated molecular patterns (DAMPs) [70]. Lipopolysaccharide (LPS) induces astroglial polarization through the release of several cytokines, including IL-1β, IFN-γ, and TNF-α, which bind to corresponding receptors on astrocytes. M1 microglia may also be activated by DAMPs-induced release of purine metabolites, HMGB1, and heat shock proteins, which activate toll-like receptors and promote cytokine secretion [71]. SE may also induce A1 and M1 polarization by directly activating receptors on astroglial surfaces [70]. The ionotropic ATP-gated P2X7 receptor is a plasma membrane receptor exclusively expressed by central nervous system (CNS) neurons and glia [72]. P2X7 is activated by LPS and ATP efflux during seizure activity, resulting in NLRP3 inflammasome activation and cytokine release [70,72,73]. Overexpression of P2X7 receptors induces a pro-inflammatory, pro-convulsant, and pharmacoresistant phenotype, whereas ablation of this receptor induces a pharmacosensitive phenotype in in vivo and in vitro SE models [73,74,75].

Following their polarization, A1 astrocytes release several neurotoxins implicated in inflammation and oxidative stress, such as TNF, IL-1β, glutamate, nitric oxide (NO), and reactive oxygen species (ROS). Additionally, A1 astrocytes can induce an iron-dependent form of regulated cell death termed ferroptosis by secreting neurotoxic long-chain saturated lipids via the chemokine receptor CXCL10/CXCR3 axis [76]. Subsequent iron dyshomestasis and the formation of ROS induce neuroinflammation in a positive-feedback loop [77]. A1 astrocytes may also affect synaptic activity by actively releasing glutamate in response to prolonged seizure activity and increased [K^+^]o [70]. The 4-AP SE model shows increased expression of the complement protein C3, a specific marker of A1 astrocytes, which promotes the expression of the transient receptor potential vanilloid type 1 (TRPV1) [78]. TRPV1 is a ligand-gated non-selective cation channel involved in several functions in the CNS, including regulating extracellular Ca^2+^ and Na^+^ influx, neuroglial proliferation, migration, and apoptosis [79]. Upregulation of TRPV1 is associated with decreased synaptic density and decreased expression of proteins essential for synaptic integrity. Ablation of the TRPV1 protein in TRPV1-knockout mice ameliorates neuronal injury seen in 4-AP-induced SE [80]. M1 microglia exert a similar neurotoxic effect through promoting the transcription of M1-associated genes, including inducible nitric oxide synthase (iNOS), cyclooxygenase-2 (COX-2), and HMGB1 [71]. Lithium-pilocarpine models show increased expression of iNOS, a primary marker of M1 microglia, associated with reduced glutamate transporter 1 expression, and increased NMDA receptor NR1 subunit expression in the hippocampus [79,81]. Hence, reducing M1 polarization via inhibition of myeloid differentiation primary response gene 88 (MyD88) has been found to increase glutamate transporter 1 expression and reduce NMDA receptor NR1 subunit expression, and subsequently ameliorate neurotoxicity [80]. Among the compounds activated by polarized neurotoxic astroglia, four in particular have been strongly implicated in the initiation of signaling cascades that exacerbate SE (IL-1β, TNF-α, COX-2, and HMGB1). These compounds can reciprocally induce A1 and M1 polarization, creating a positive feedback loop [38,70,82,83]. The P2X7 receptor is primarily responsible for the release of the neurotoxic cytokine, IL-1β, from glial cells and neurons [75]. IL-1β phosphorylates the NR2B subunit of the NMDA receptor via IL-1RI-dependent activation of Src kinases and sphingomyelinases, thereby increasing NMDA receptor expression in post-synaptic neurons [82,84]. Additionally, IL-1β facilitates TNF-α release, thereby promoting glutamate exocytosis from astrocytes and upregulating AMPA receptors while simultaneously reducing GABA receptor expression [82]. Furthermore, IL-1β contributes indirectly to the inhibition of glutamate uptake via induction of NO production [82,85]. The aforementioned ictogenic effects of IL-1β are mediated by the IL-1 receptor 1 (IL-1R1)/Toll-like receptor (TLR) signaling in glia and neurons [85,86]. IL-1β has been found to significantly decrease the GABA_A_ evoked currents in brain slices from various in vivo models as well as patients with temporal lobe epilepsy [83,87]. Additionally, IL-1β has been found to increase neuronal apoptosis following lithium-pilocarpine-induced SE in a mechanism independent of IL-1RI activation [88]. In chemically and electrically induced SE models, antagonism of IL-1 receptor, using competitive IL-1 receptor type 1 antagonist, and inhibitors of IL-1β cleavage and release, has been found to delay seizure onset, reduce the damage of the blood–brain barrier (BBB), and reduce neuronal loss in the forebrain [64,87] though it did not seem to alter the risk for future spontaneous recurrent seizures [86]. Neuronal injury is amplified by the concurrent release of other pro-inflammatory cytokines such as HMGB1, which is also implicated in activating the IL-1R1/TLR signaling cascade [70,86]. Upregulation of HMGB1 may be involved in the pharmacoresistance of SE, whereas its inhibition has been found to extend the therapeutic window in the KA-SE model [89]. Another molecule heavily implicated in the propagation of SE is the prostaglandin COX-2, which is released by hyperexcitable neurons and M1 microglia [4,90]. Upregulation of COX-2 in hippocampal neurons is evident in the early hours of SE onset, far preceding neuronal death in in vivo and in vitro SE models, suggesting that COX-2-mediated inflammation is a reaction to seizure activity rather than neuronal death [91,92]. Additionally, antagonism of COX-2 in vitro and in chemically induced SE using neuron-specific conditional knockout mice ameliorates excitotoxic neuronal death [92,93], whereas antagonism of glutamate receptors (AMPA, NMDA, and metabotropic) inhibits in vitro COX-2 mRNA expression [92]. This suggests that the upregulation of COX-2 receptors in SE stimulates the release of glutamate, which in turn stimulates the expression of COX-2 mRNA in a positive feedback loop [90,92]. However, COX-2 inhibition had no observable benefit on neuronal death in electrically induced SE [94] or on epileptogenesis in the pilocarpine model [95]. This may be related to the timing of COX-2 inhibition, where studies that administered prophylactic treatment found a neuroprotective effect [92,93,95], whilst the study that administered COX-2 inhibitors in the latent period, after SE termination, did not [93]. Notably, both COX-2 and HMGB-1 are implicated in the upregulation of P-glycoprotein at the BBB, a drug efflux transporter associated with pharmacoresistance, through the regulation of JNK signaling by COX-2 and RAGE/NF-κB signaling by HMGB-1, respectively [96,97,98]. Upregulation of P-glycoprotein has been reported in drug-resistant epilepsy [99] and SE models and is associated with reduced efficacy of ASMs that are substrates for P-glycoprotein, namely levetiracetam, phenytoin, and phenobarbital [100,101]. TNF-α is yet another cytokine heavily implicated in SE pathogenesis as it controls neural cadherin (N-cadherin), an essential protein for the development of excitatory and inhibitory synapses. TNF-α boosts excitatory transmission by upregulation of glutaminase enzyme at gap junctions, enhancing the expression of AMPA receptors while triggering GABA receptor endocytosis [82]. In pilocarpine models, TNF-α has been found to induce endothelin-1 (ET-1) release and expression in neurons and endothelial cells, resulting in nitric oxide synthase (eNOS) activation in endothelial cells and a subsequent increase in BBB permeability [102]. BBB disruption secondary to neuroinflammation and energy depletion permits the infiltration of immune cells into the brain, further propagating the production of inflammatory mediators [38,103]. Experimental BBB disruption in models of osmotic opening of the BBB has been found to be a sufficient trigger of seizures even in the absence of CNS pathologies [104]. SE patients are reported to have a high cerebrospinal fluid (CSF)/serum albumin ratio (QAlb), a measure of the integrity of the BBB, compared to healthy controls [105]. Additionally, histologic studies show albumin accumulation in human epileptic brain tissue [106]. This is thought to occur within an hour of SE onset and to progress over the course of sustained seizure activity; using sodium fluorescein and Evans blue dye, the BBB of in vivo SE models shows permeability to micromolecules within an hour and to macromolecules in the following hours [107,108]. Serum proteins, including albumin, and leukocytes can leak into the brain through a disrupted BBB. Subsequently, albumin binds to astrocytic TGF-β receptors, reducing the expression of Kir 4.1 channels and glutamate transporters, and initiating a cascade that enhances neuronal hyperexcitability and astroglial polarization [109]. Astroglial polarization is further increased due to increased iron accumulation within astroglia through leakage of iron-rich blood components across a broken BBB [77]. Additionally, as lymphocytes infiltrate the brain parenchyma, they are activated to produce pro-inflammatory cytokines, sparking yet another positive feedback loop [82]. Additionally, the breakdown of the BBB exposes the brain to coagulation factors such as serine protease thrombin, which potentiates SE through the activation of protease-activated receptor 1 (PAR1) [110]. Moreover, the breakdown of the BBB is associated with increased expression of the losartan-sensitive angiotensin II receptor (AT1). Upregulation of this receptor can compromise cerebral blood flow through myogenic vasoconstriction. Blockade of the AT1 receptor with losartan was associated with improved cerebral perfusion and neuroprotection in chemically induced models [111,112]. Additionally, losartan appears to be neuroprotective through the inhibition of AT1-activated NAD(P)H oxidase-dependent ROS, suppression of microglia-mediated inflammatory responses, and reduction in cellular edema via normalization of aquaporin-4 (AQP4) expression [111,112]. BBB disruption in SE is likely the result of activation of matrix metalloproteinases (MMPs) and subsequent degradation of the basal lamina and tight junction proteins and pericyte detachment [113]. Experimental inhibition of MMP-9 and MPP-12 in SE models has been found to limit myeloid infiltrates and albumin extravasation and to reduce seizure duration and the frequency of spontaneous recurrent seizures, but its effect on neuronal death has been variable, as some studies found that it was neuroprotective [114,115], but others did not [116].

### 4.3. Altered Energy Homeostasis

Under physiological conditions, brain tissues are nearly 10-fold more metabolically active than other tissues, accounting for 20% of the body’s energy consumption despite limited brain energy reserves [117,118]. This is achieved by neurometabolic coupling, which ensures that utilization of energy substrates parallels neuronal activity [119]. Ionic pumping required for excessive neuronal discharge during SE significantly increases cerebral demands for energy substrates. A homeostatic response to this significant increase in cerebral metabolic rate is a simultaneous increase in cerebral blood flow termed “neurovascular coupling”, facilitating the transportation of metabolic substrates required for “neurometabolic coupling” [46,103]. However, prolonged seizure activity abolishes physiological autoregulation of cerebral blood flow (CBF), making it blood-pressure dependent. Cerebral perfusion is maintained at the start of seizure activity, where systemic blood pressure is extremely elevated. However, systemic blood pressure eventually drops as a result of decreased sensitivity of blood vessels to catecholamines due to lactic acidosis and failure of catecholamine release. In GCSE, CBF is estimated to fall within 30–60 min of SE onset, thereby reducing the supply of O_2_ and glucose and a subsequent drop in ATP and phosphocreatine associated with disrupted protein synthesis [46,52]. This affects the expression of transporters at the BBB, thereby reducing the uptake of nutrients into the brain and exacerbating waste product buildup [113]. Furthermore, neuronal depolarization may be associated with cerebral vasoconstriction termed “inverse coupling”. This is partly due to enhanced synthesis of vasoactive prostaglandins during seizures and increased Ca^2+^ in vascular smooth muscle [120,121]. Subsequent energy failure affects active transporters required for neuronal repolarization, leading to prolonged neuronal depolarization and sustainability of seizure activity [103].

Additionally, diminished phosphate stores prevent glutamate reuptake with subsequent activation of NMDA receptors and AMPA receptors, resulting in glutamate neurotoxicity and excessive Ca^2+^ influx into the mitochondria, thereby disrupting mitochondrial function and ATP production, further exacerbating energy failure [35,122]. Energy failure leads to inhibition of the Na^+^/K^+^ pump and Ca^2+^ pump, thereby maintaining activation of the NMDA receptor and increasing [Ca^2+^]i. Ultimately, disruption of the Krebs cycle reduces the availability of glutamate and subsequently GABA [38,52]. Additionally, ATP depletion and subsequent fall in GTP levels increase GDP levels, leading to GDP displacing GTP from its binding site and interrupting protein synthesis. Interruption of the energy-consuming protein synthesis may divert some of the energy to neuronal survival and may interfere with synaptic potentiation [46]. Although metabolic deficit would hypothetically limit the progression of seizures, either due to depletion of glutamate synthesis in the epileptic focus or by the neuronal death of the epileptic focus, or indeed both [53,123]. ATP depletion and lactate accumulation in such a metabolically demanding state induce necrosis, which may be visible on perfusion-weighted magnetic resonance imaging (PWI) [52,124]. In animal models of CSE, neuronal injury spanning the cortex, cerebellum, and hippocampus has been documented in a pattern similar to that seen in cardiac arrest or severe hypoglycemia [9].

### 4.4. Excitotoxicity

In the first 24 h of chemically induced SE, canine models show elevation of CSF glutamate and reduction in CSF GABA [53]. In this acute phase, there is increased synthesis and efflux of glutamate in cortical and hippocampal synaptosomes [53,125]. Glutamate activates NMDA receptors, AMPA receptors, and kainate receptors to open their associated ion channels, permitting the influx of Ca^2+^ and Na^+^ ions. However, excessive synaptic release of glutamate and subsequent sequestration of Ca^2+^ into the cell can overwhelm neuronal Ca^2+^-regulatory mechanisms [19,126]. Although glutamate receptor hyperactivation results in excessive influx of Na^+^ as well, it is the altered Ca^2+^ dynamics that mediate glutamate neurotoxicity. It is suggested that excitotoxicity has an acute Na^+^-mediated phase characterized by swelling and a delayed Ca^2+^-mediated phase characterized by neurodegeneration [126]. Controlled in vivo models of glutamate hyperactivation with exclusive Na^+^ influx show that whilst excessive Na^+^ influx causes toxic and sometimes irreversible cellular swelling, it does not mediate a neurodegenerative cascade. On the other hand, glutamate hyperactivation with exclusive Ca^2+^ influx results in swelling and delayed neuronal degeneration [52,126]. In fact, Ca^2+^ influx alone, by using the Ca^2+^ ionophore A23187, has been shown to induce apoptosis in rat cortical cultures irrespective of glutamate receptor hyperactivation [127]. Notably, intracellular Na^+^ accumulation can perpetuate Ca^2+^ overload as it reverses the activity of the Na^+^/Ca^2+^ exchanger, resulting in Ca^2+^ accumulation [126]. Ca^2+^ overload is further exacerbated by Ca^2+^ release from the endoplasmic reticulum (ER) secondary to metabotropic glutamate receptor (mGluR) activation, reduced activity of sarco/ER calcium ATPase (SERCA), and Ca^2+^-induced Ca^2+^ release [128,129]. CA^2+^ overload is largely considered an irreversible step in the pathogenesis of neurodegeneration [130]. As an important second messenger, aberrant Ca^2+^ homeostasis can alter gene expression, neurotransmission, and neuroplasticity [127]. The excessive influx of Ca^2+^ into the cell results in its sequestration within the cytoplasm and cellular organelles, most notably within the mitochondria [38,126]. The sequestration of Ca^2+^ within the mitochondrial matrix activates the mitochondrial permeability transition pore to permit the efflux of Ca^2+^ into the cytosol. Opening of the mitochondrial permeability transition pore causes hyperpermeability to solutes and low molecular weight substances and mitochondrial membrane depolarization, which becomes irreversible in conditions of oxidative or nitrosative stress [131]. Mitochondrial membrane depolarization disrupts the mitochondrial electron transport chain, causing electron leakage from the electron transport chain, decreased oxygen consumption, and ATP depletion [131]. ATP depletion disrupts the function of ATP-dependent Ca^2+^ pumps, further exacerbating Ca^2+^ overload [132]. Additionally, Ca^2+^ overload activates Ca^2+^-dependent enzymes, including calpains, proteases, endonucleases, and lipases, protein kinase C, calmodulin-dependent kinases, NOS, and endonucleases, resulting in direct structural or indirect functional damage by altering phosphorylation or via free radicals [128,130]. Furthermore, excessive Ca^2+^ influx upregulates the Ca^2+^/calmodulin dependent phosphatase calcineurin, which in turn activates cofilin, thereby triggering the depolymerization of actin cytoskeleton (F-actin) and the loss of dendritic spines [133]. In the CNS, F-actin is a major synaptic protein involved in the structural support and plasticity of the dendritic spines, as well as in the translocation and anchorage of postsynaptic receptors [133]. Neuronal cytoskeletal alterations can affect the ability to transmit information and cargo between cell organelles and synapses, causing a dying-forward process, seen in neurodegeneration [134]. Experimental disassembly of F-actin using lartrunculin A has been found to upregulate extracellular excitatory neurotransmitters, leading to long-term changes in neuronal excitability, and the emergence of sporadic spontaneous seizures in treated rats [135], which can be prevented by administration of the actin filament stabilizer ascomicin [136].

Notably, Ca^2+^-mediated neurotoxicity appears to be “source specific” as Ca^2+^ influx through L-type voltage-sensitive Ca^2+^ channels is not as neurotoxic as Ca^2+^ influx through NMDA receptors and AMPA receptors, suggesting the possibility of yet to be known ion-receptor interactions [128,132,137]. In vivo models of SE show decreased expression of mRNA encoding Glu receptor 2 (GluR2), the subunit that limits Ca^2+^ permeability of AMPA receptors, in the hippocampal subfields cornu ammonis (CA)1, CA3, and CA4, while the expression of GluR2 is unchanged in granule cells of the DG. The latter are significantly more resistant to excitotoxic death than CA1 and CA3, highlighting the significance of Ca^2+^ permeability in excitotoxicity [138,139]. Indeed, inhibition of GluR2 via intracerebroventricular injection of specific GluR2 antisense oligonucleotides has been found to cause neurodegeneration in CA1 and CA3, suggesting that downregulation of GluR2 and subsequent hyperpermeability of AMPA receptors mediate neurodegeneration [140]. Conversely, the expression of other AMPA receptor subunits (GluA1, GluA3, and GluA4) is either unchanged or increased. These changes are concurrent with a reduction in the expression of GABA_A_ mRNA in the areas most susceptible to neuronal injury, suggesting that cell selectivity of neuronal death is correlated to cell selectivity of excitotoxicity [139]. In chemically induced SE, after 72 h of SE onset in canines, both CSF glutamate and GABA start to drop below normal levels and only return to normal levels after 2 months, likely due to limited synthesis. Spontaneous seizure termination is also seen after 72 h of seizure induction, coinciding with, though not necessarily caused by, neuronal death of the epileptic focus. Whether SE termination is induced by depletion of glutamate synthesis or by the neuronal death of the epileptic focus remains to be elucidated [53], though it is likely that the latter hypothesis is more accurate, as surviving neurons that have been permeable to Ca^2+^ sustain elevated [Ca^2+^]i for days and weeks. Through second messenger effects, post-SE Ca^2+^ plateau results in alterations in plasticity, in Ca^2+^ homeostatic mechanisms, gene expression, and transcription of proapoptotic and/or epileptogenic genes, thereby serving as a substrate for epileptogenesis [19,141].

### 4.5. Oxidative Stress and Mitochondrial Dysfunction

The metabolic rate of the brain and, thus, its production of ROS is exponentially elevated during seizure activity, far exceeding its antioxidant capacity, resulting in oxidative stress [142]. The brain is particularly susceptible to oxidative stress due to its high content of polyunsaturated fatty acids and the chemical reactivity of neuroglial fatty acids, as well as its limited antioxidant defense capability [142,143]. SE is associated with high cerebral expression of F2-isoprostanes and malondialdehyde, products of lipid peroxidation, and high expression of 8-hydroxy-2-deoxyguanosine, a product of DNA oxidation [144,145,146]. Conversely, the cerebral expression of the antioxidants aconitase and glutathione is reduced in hippocampal tissues retrieved from SE models [147]. Additionally, there is a correlation between the expression of ROS in SE models and the extent of neuroinflammation and neuronal death [148]. In SE, ROS are predominantly generated by nicotinamide adenine dinucleotide phosphate (NADPH) oxidase (NOX) following Ca^2+^ entry through activated NMDA receptors. Reactive nitrogen species (RNS) are also exacerbated by excessive Ca^2+^ entry, which activates NOS [142]. NO not only acts as a retrograde messenger, inducing glutamate release from presynaptic terminals and mediating the action of glutamate in stimulating cyclic GMP concentrations, but can also react with ROS to form peroxynitrite, which is a highly reactive oxidant [38,52,126,149]. Through oxidation and nitration processes, peroxynitrite mediates several cytotoxic effects, including protein nitration and oxidation, lipid peroxidation, ion channel inactivation, DNA damage, activation of MMPs, interference with redox signaling, and inactivation of enzymes [150]. DNA damage triggers the activation of poly(ADP-ribose) polymerase-1 (PARP-1), which cleaves NAD(+) into nicotinamide and ADP-ribose. PARP-1 then polymerizes ADP-ribose on nuclear acceptor protein, thereby depleting NADH and dampening ATP production [150]. Additionally, proteins that have undergone peroxynitrite modification are targeted by the ubiquitin-proteasome system for clearance [150]. The mitochondria and mitochondrial enzymes are particularly susceptible to peroxynitrite toxicity, as it can react with several key components of mitochondrial function [38,126]. Peroxynitrite and NO inhibit the complexes of the electron transport chain through cysteine oxidation, tyrosine nitration, and the degradation of iron sulfur clusters [150]. Oxidative and nitrosative stress decrease the threshold for mitochondrial permeability transition pore opening, thereby potentiating Ca^2+^-mediated mitochondrial membrane depolarization, electron leakage, and subsequent ATP depletion [131]. ATP depletion is further exacerbated by peroxynitrite-dependent inhibition of the mitochondrial tricarboxylic acid cycle enzyme aconitase and acceleration of ADP-ribose turnover [142,150]. Additionally, Peroxynitrite inactivates nicotinamide nucleotide transhydrogenase, thereby depleting NADPH and subsequently preventing mitochondrial regeneration of glutathione, further amplifying oxidative stress within the mitochondria [150]. Sustained opening of mitochondrial permeability transition pore not only disrupts the mitochondrial membrane potential but also permits cytochrome c release into the cytosol, thereby activating pro-apoptotic caspases [142,148]. Exaggerated opening of mitochondrial permeability transition pore in the presence of PARP overactivation triggers necrosis [150]. In addition to apoptosis and necrosis, oxidative and nitrosative stress induce ferroptosis [151]. Ferroptosis triggers a cycle in which oxidative stress activates the inflammasome and induces the expression of pro-inflammatory cytokines, such as COX-2 and HMGB1, which in turn fosters the generation of more ROS [148,152,153]. Additionally, ROS help stabilize HMGB1 in its disulfide isoform, which mediates pro-inflammatory cytokine secretion through TLR4 [153,154]. Experimental use of antioxidants has been shown to reduce the expression of TNF-α and HMGB, lower ferritin levels and ameliorate neurodegeneration, and reduce the risk of spontaneous recurrent seizures in chemically induced SE models [146,151,153,155,156,157]. Although mitochondria generate ROS secondary to electron leakage, Ca^2+^ ions competing with cytochrome c for binding sites on cardiolipin, and peroxynitrite-mediated inhibition of manganese superoxide dismutase (MnSOD) [131,150], their contribution to seizure-related oxidative stress is negligible in the grand scheme due to the aforementioned chain of events that ultimately lead to mitochondrial failure [38,126]. Indeed, the bulk of ROS is generated from other sources such as NOX, xanthine oxidase, monoaminoxidase, cyclooxygenase, and lipoxygenase [142].

### 4.6. Neuronal Death

SE is associated with elevated biofluid levels of neuron-specific enolase (NSE), neurofilament heavy chain (NfH), Neurofilament light chain (NfL), and tau-protein [2,17,105,158,159,160]. Additionally, postmortem analysis of post-refractory SE reveals widespread brain atrophy, particularly in the hippocampus and the entorhinal cortex [4,35]. This is true even for NCSE, though the extent of brain damage may vary [4,161]. Neuronal death pathways are activated directly by mitochondrial dysfunction or indirectly via excitotoxicity and ATP depletion-related mechanisms [61,122,162,163]. Neuronal death caused by SE exhibits characteristics of necrosis and apoptosis [61,164]; the processes of which are detailed in previous articles [122,163]. Excitotoxicity and subsequent Ca^2+^ overload activate calpains that promote apoptosis or necrosis by lysosomal cell death [38,52,61,122,162,163]. Additionally, SE causes underexpression of Bcl-2, a protein that normally blocks the proapoptotic Bax protein, leading to a reduced Bcl-2/Bax ratio. However, in chemically induced models, this alteration is only evident one to three days after seizure induction and normalizes from thereon, suggesting upstream regulation of apoptosis [61]. Ferroptotic death has recently been recognized as one of the forms of neuronal death in SE. It is a form of regulated cell death that occurs as a consequence of iron-catalyzed lipid peroxidation [151]. Fluoro-Jade staining shows that neuronal death is not homogenous across all neuronal cell populations in SE models [164]. Selective neuronal vulnerability, a feature of neurodegenerative processes, is seen in SE [165,166]. In the hippocampus, neurodegeneration is most notable in the hilus and pyramidal cell subfields CA1 and CA3 24 h after chemical induction of SE, while GABAergic interneurons in CA1 and CA3, CA2 neurons, and DG cells are preserved [20,138,164]. The mechanisms behind the differential susceptibility of different populations are not clear, although recent evidence implicates differential expression of NMDRs, AMPA receptors, and Ca^2+^-binding proteins [130,166]. It is suggested that the vulnerability of CA1 pyramidal neurons to neurodegeneration is linked to their high content of NMDA and AMPA receptors and subsequent high Ca^2+^ permeability, while the vulnerability of hilar GABAergic interneurons may be related to their high dependency on aerobic metabolism [43,130]. The relative immunity of DG cells and pyramidal neurons in CA2 is attributed to the high density of calcium-buffering proteins such as chromogranin A and parvalbumin [130]. In addition to neuronal populations, differential astrocytic function may be similarly implicated in selective neuronal vulnerability, as CA1 astrocytes have been found to have less density of astrocyte-specific glutamate transporter (GLT-1) compared to DG astrocytes in models of transient forebrain ischemia [61]; whether a similar effect is induced by SE is worth investigating.

## 5. Mechanisms of Epileptogenesis

SE is an epileptogenic event that increases the risk of developing future epilepsy by threefold compared to single seizures, and with 20–40% of SE survivors developing acquired epileptogenesis [19,35]. Epileptogenesis refers to the molecular and cellular alterations causing the development and/or expansion of tissue capable of triggering spontaneous epileptic seizures following neurological injury [167,168]. It is a progressive process with several mechanisms culminating over a latent period in the emergence of spontaneous recurrent seizures. The latent period of epileptogenesis is a seizure-free state of epileptic maturation characterized by functional and structural alterations over an indeterminate period of time [168]. The duration of the latent period is different among studies; while most studies suggest that epileptic tissue matures over several days to weeks [49,164,169], others argue that epileptogenesis is evident immediately after SE, albeit subclinical [168,170,171]. It is hypothesized that the extent of neuronal injury is inversely related to the latent period, i.e., seizures with extensive neuronal loss are associated with a short latent period and faster epileptic maturation [168]. Epileptic maturation involves several mechanisms, including neurogenesis, axonal and dendritic reorganization, neuroinflammation, breakdown of the BBB, angiogenesis, and acquired channelopathies [82]. Despite the widespread nature of lesions seen in models of SE-induced epilepsy, the sum of epileptogenesis studies is based on analysis of the limbic area in SE models [54], as is the literature reviewed in this section.

### 5.1. Aberrant Neurogenesis

Although CNS neurons are terminally differentiated post-mitotic cells, neurogenesis and synaptogenesis of the DG continue throughout life [172]. Emerging evidence suggests that neurogenesis may also extend to other parts of the hippocampal formation as well as other limbic structures [173]. In dentate neurogenesis, neurons are produced from proliferating stem/progenitor cells in the subgranular zone (SGZ) of the DG, from which they migrate to incorporate into the hippocampal circuitry. Dentate neurogenesis is activity-dependent and appears to be enhanced by brain/hippocampal injury, as in SE [174]. SE is associated with increased CSF levels of progranulin, a biomarker of neurogenesis, in both SE models and patients [175]. However, SE-associated neurogenesis may be misdirected, as most of the new neurons develop ectopically in the hilar region of the hippocampal formation, potentially leading to the formation of faulty and epileptogenic circuitry in the injured hippocampus [12,52]. Monitoring dentate neurogenesis in vivo using pulsed injections of 5′-bromodeoxyuridine (BrdU) [169,176,177] and retroviral labeling of dividing cells and their progeny [178], or in vitro using doublecortin (DCX) [174,177] reveals that SE is associated with chronic enhancement of neurogenesis that can persist for months following chemically and electrically induced SE. Additionally, the size of the hilar ectopic granule cell (EGC) population is reportedly positively correlated with the frequency of spontaneous recurrent seizures [179] and inversely correlated with the time to spontaneous recurrent seizures emergence [174]. Topologically, hilar EGC dendrites are more branched, tortuous, and entrenched along the longitudinal and horizontal DG axes, thereby permitting convergent afferent input and divergent efferent output. Additionally, projections from hilar EGCs promote mossy fiber innervation of CA3 as well as abnormal mossy fiber sprouting [180]. Seizure-induced neurogenesis is stimulated by neuroinflammation and brain-derived neurotrophic factor (BDNF) signaling [4,181]. Following neuronal injury, activated microglia are concentrated at the site of injury where they stimulate the migration and proliferation of neural CNS precursor cells via a number of pro-inflammatory mediators, including COX-2 [182,183]. Additionally, SE is associated with a prolonged overexpression of BDNF and its receptor tropomyosin-related kinase receptor type B (TrkB) [21]. Increased BDNF/TrkB signaling increases glutamate release through the activation of PLCγ-mediated Ca^2+^ release, which also stimulates BDNF production in a positive feedback loop. Additionally, TrkB and VEGF promote the stimulation of angiogenesis, resulting in increased vessel density within the hippocampus [184]. However, the newly formed vessels are tortuous and leaky, resulting in BBB disruption, downregulation of tight junctions, and subsequent extravasation of ions and proteins, thereby triggering neuroinflammation and epileptogenesis [185]. Additionally, epileptogenesis is linked to differential expression of genes associated with neurogenesis in the hippocampus, such as Notch1-201, which, when targeted, can ameliorate neurogenesis and epileptogenesis [186,187]. Whether neurogenesis is epileptogenic by nature or by association with other epileptogenic factors has not been established [12,171]. Ultimately, the fate of the newly developed neurons is debatable, while some studies suggest that most neurons are not differentiated into mature neurons [169], others suggest that they are [177].

### 5.2. Mossy Fiber Sprouting

Mossy fibers are glutamatergic, efferent axons of DG that course through the dentate hilus and stratum lucidum, and innervate hilar cells and CA3 pyramidal cells. They directly excite CA3 pyramidal neurons and indirectly inhibit CA3 pyramidal neurons by activating GABAergic interneurons. This permits both feedback and feedforward inhibition of DG cell output [180]. Mossy fiber sprouting (MFS) is the growth of new axon collaterals that extend beyond the dentate hilus, crossing the granular cell layer into the inner molecular layer of DG, where they synapse with dendrites of other granule neurons, forming monosynaptic interconnections. These interconnections form aberrant positive-feedback loops to the granule cells themselves that do not normally exist, thereby increasing network synchronization and propagation of seizures, as illustrated in Figure 2 [38,43]. Additionally, these interconnections create a homogenous neural circuit, which has been shown by patch clamp recordings from human cortical pyramidal neurons to harbor ictogenesis [188]. Mossy fibers may also sprout into CA3 and into CA2 and CA1, increasing connectivity to pyramidal cells and extending the divergence of the mossy fiber dendrites along the septotemporal axis of the hippocampus [12,20,54,171]. The “mossy fiber” theory attributes hippocampal hyperexcitability to MFS and the formation of new excitatory circuits as a result. The “mossy fiber” theory is supported by evidence of mossy fiber sprouting in the dentate gyri of chemical and electrical SE models [12,168]. An alternative theory, termed the “dormant basket cell” theory, attributes hippocampal hyperexcitability to deafferentation of basket cells following mossy cell death. These mechanisms are not mutually exclusive and may, in fact, work synergistically to tilt the balance toward hyperexcitability [37]. Although MFS precedes the occurrence of spontaneous recurrent seizures in electrical and chemical SE models and is evident in postmortem tissue from patients with temporal lobe epilepsy, its role in epileptogenesis is still the subject of ongoing research [12,171]. MFS peaks at 2 months after electrical stimulation of SE as per increased Timm staining in the inner molecular layer of the DG [49]. The extent of sprouting is correlated with the severity of SE and the extent of neuronal death [20]. Although MFS is associated with a concordant increase in NMDA receptor density [189] and bursts of action potentials and spontaneous after-discharges in response to antidromic stimulation [190], there does not appear to be a correlation between the extent of sprouting and the frequency of spontaneous recurrent seizures [164]. The main mechanisms behind MFS include activation of the mTOR signaling pathway and aberrant homeostatic synaptic plasticity [43,191]. The mTOR is a ubiquitously expressed serine/threonine kinase that regulates several intracellular processes in response to a variety of extracellular signals, including glutamate and BDNF [191]. In the CNS, the mTOR pathway regulates neurite growth, dendritic arborization, synaptic plasticity, synaptogenesis, and ion channel expression, likely by influencing protein translation or Akt activity [192,193]. In *TSC*/*PTEN* genetic models, mTOR signaling increases evoked synaptic responses, axonal and dendritic growth, the density of synaptic vesicles, and synaptic density in both excitatory and inhibitory neurons. Additionally, the mTOR pathway is involved in the modulation of glial activation and proliferation [192]. Hyperactivation of the mTOR pathway produces epilepsy in animal models [194], primarily through autophagy failure and its subsequent excitatory effect on neurotransmission [195]. In keeping with this, inhibition of mTOR by rapamycin attenuates MFS in *TSC*/*PTEN* genetic models [196] and chemically induced SE models [183] and has been shown to suppress spontaneous recurrent seizures [196]. In addition to mTOR-mediated structural changes, hyperactivation of mTOR has been shown to impair mGluR-long-term depression (LTD) and enhance long-term potentiation (LTP) at hippocampal synapses [194]. Additionally, mTOR hyperactivation has been linked to decreased frequency and amplitude of miniature inhibitory post-synaptic currents (mIPSCs) and of evoked IPSCs, resulting in increased excitability via reduced inhibition [194]. The mTOR pathway is activated within 30 min to 2 h of SE induction in chemical SE models [162,193]. Administration of rapamycin in SE models has been shown to reduce seizure activity, ameliorate MFS, and reduce spontaneous recurrent seizures [193,197,198]. However, whether this is solely attributed to blocking mTOR signaling or is at least partially affected by the improved BBB integrity seen in those models is yet to be elucidated [197]. Similarly, mTOR inhibition might restore autophagy-dependent internalization of glutamate receptors, thereby alleviating excitotoxicity-mediated Ca^2+^-overload [195]. Therefore, the anti-ictogenic effect of mTOR inhibition and inhibition of MFS may be influenced by concurrent changes, including alterations in the release of neurotransmitters and the neuro-immune interactions [38,199].

### 5.3. Neuronal Death

Neuronal death is not only correlated with neurogenesis and MFS; it is the strongest predictor of epileptogenesis post-SE [20]. It is argued that post-SE structural reorganization is not sufficient to cause epilepsy, as they are evident prior to the emergence of spontaneous recurrent seizures. Rather, neuronal loss is thought to trigger secondary processes that further increase network excitability on top of granule cell disinhibition [168]. Researchers proposed the “recapitulation of development” hypothesis to explain the link between neuronal injury and synaptic potentiation. The hypothesis suggests that loss of synaptic input from dying neurons triggers a signal to induce a number of morphological and functional alterations [4]. Neuronal activity is maintained within a physiological range by virtue of homeostatic synaptic plasticity (HSP), which is an activity-dependent negative feedback mechanism that prevents excessive excitation or inhibition. However, in cases of loss of synaptic input and prolonged network silence, HSP fails to regulate synaptic strength, and maladaptive homeostatic plasticity takes over, offsetting a cascade of epileptogenic alterations [200,201]. Initial network silence triggers compensatory increase in neurotransmitter release and upregulation of Na^+^ channels, excitatory postsynaptic receptors, and downregulation of synaptic GABAA receptors [43]. MFS appears to follow mossy cell death and deafferentation of basket cells in the hilus as a compensatory mechanism in response to loss of their target cells or afferents [20,164]. These morphological and functional alterations are reminiscent of the mechanisms of deafferentation-induced epileptogenesis in cortical undercut models and chronic GABA infusion models [43,201]. In cortical undercut models, the membrane potential of neurons in the undercut cortex is more hyperpolarized compared to surrounding neurons due to increased [K^+^]o secondary to neuronal injury. This change in electric polarity across the undercut cortex and the surrounding areas connected to it creates conditions that facilitate the generation of seizures at the border between the undercut and intact cortex, which then propagate outward into normal tissue [43,200]. A second hypothesis, termed the “neuronal death pathway” hypothesis, has been proposed to explain the close link between neuronal death and epileptogenesis. The neuronal death pathway hypothesis states that molecular mechanisms underlying neurodegenerative pathways are at least partially responsible for seizure-induced epileptogenesis, most notably altered Ca^2+^ homeostasis and its subsequent effect on plasticity, gene expression, and neurotransmitter release [4,19]. Indeed, chelation of [Ca^2+^]i in neuronal culture models of epilepsy can prevent spontaneous recurrent seizures, especially when combined with NMDA receptor antagonists [137].

## 6. Discussion

SE is a time-sensitive condition resulting from the failure of the mechanisms responsible for seizure termination and the initiation of mechanisms that sustain seizure activity, with a myriad of pathological changes that accumulate over time (summarized in Figure 3), causing long-term sequela, including neuronal death and alteration of neuronal networks [12,38]. The transition from single seizures to SE is established by early maladaptive internalization of synaptic benzodiazepine-sensitive GABA_A_ receptors and increased recruitment of NMDA receptors, AMPA receptors, as well as benzodiazepine-insensitive GABA_A_ receptors at synapses. Receptor trafficking progresses through the established phase of SE, causing time-dependent resistance to therapies involving modulation of benzodiazepine-sensitive GABA_A_ receptors [48,52]. Although guidelines recommend first-line treatment with benzodiazepines, most patients are in the established stage of SE by the time they are presented to a hospital [46,202]. At which point, GABA_A_ agonists would not be able to fully restore inhibition as the number of synaptically active GABA_A_ receptors is significantly reduced [12]. In SE models, increased synaptic localization of NMDA receptors modulates AMPA externalization and GABA_A_ internalization; early administration of NMDA receptor antagonists such as ketamine not only targets glutamate hyperactivation but also suppresses the internalization of GABA_A_ receptors [87]. Accumulating experimental evidence suggests that simultaneous polytherapy targeting both the failure of inhibition and the runaway excitation is far more effective than sequential polytherapy targeting one aspect of the pathogenesis at a time [23,46,203]. In pilocarpine models, combinations of diazepam-ketamine-valproate and midazolam-fosphenytoin-valproate were more effective than triple-dose monotherapy using the same drugs [204,205]. The combination of diazepam-ketamine-valproate was more effective than midazolam-fosphenytoin-valproate [206], likely due to the former combination targeting more aspects of the pathophysiology. Simultaneous polytherapy improves the therapeutic index, providing synergistic efficacy without compromising the safety profile, as the drug toxicity is additive [203,204,205,206,207]. Extrasynaptic GABA_A_ receptors are increasingly recognized as a potential adjunct therapy using allopregnanolone and ganaxolone, neurosteroids that modulate synaptic and extrasynaptic GABA_A_ receptors with favorable safety profiles [208,209,210]. Allopregnanolone was successful in weaning patients with super-refractory SE from general anesthesia in a phase I/II clinical trial [209] and is currently being studied in an upcoming phase III clinical trial [211]. Similarly, adjunctive galaxone has been shown to mitigate the need for anesthesia for refractory SE in a phase II trial [210] and to permit weaning of anesthesia in super-refractory SE [212]. Notably, AMPA antagonists demonstrated efficacy in chemical and electrical SE models [213,214], albeit transiently, as the seizures reappeared a few hours later, highlighting the need for continuous monitoring [214]. Seizure reemergence can alternatively be explained by the fact that AMPA externalization is NMDA-dependent and therefore targeting AMPA manages the consequence, not the cause [88]. Given the time-dependency of receptor trafficking in SE and the fact that most seizure emergencies take place outside the hospital, non-intravenous rescue therapies, such as rectal diazepam, intranasal or buccal midazolam, should be prescribed for people with a history of uncontrolled seizure disorder [215]. It is also worth investigating the safety and efficacy of non-intravenous formulations of non-BZD ASMs that are being developed for seizure emergencies, such as allopregnanolone [216] and brivaracetam [217], which may have the ability to overcome BZD pharmacoresistance, especially in cases where access to a hospital is delayed. SE activates astrocytes and microglia, which exacerbates neuronal injury through the release of cytotoxic mediators [71,82]. Subsequently, neuronal injury serves as an endogenous signal for astroglial activation in a positive feedback loop [71,80,133]. Neurodegeneration has been shown to be most extensive at 72 h following SE induction and is sustained for 30 days, in correlation with neuroinflammation [218]. Additionally, inflammatory mediators released by astroglia have been found to facilitate seizure progression, most notably IL-1β, which has been found to reduce GABA-mediated neurotransmission [84]. Additionally, neuroinflammation is associated with upregulation of P-Glycoprotein at the BBB through COX-2 and HMGB1 signaling [96,97,98]. The P-glycoprotein transporter is not only implicated in multi-drug resistance but has also been linked to endoplasmic reticulum stress and iron dysregulation, resulting in ferroptosis [96]. Targeting P-glycoprotein either directly or through COX-2 inhibition promotes entry of phenytoin and phenobarbital into the brains of animal models [98,99,100,219]. Administration of verapamil, a non-selective P-glycoprotein inhibitor, has been suggested by some researchers as a sensitizer to ASM in drug-resistant epilepsy and SE [220], but others argued against it due to possible inhibition of ASM [99]. Upregulation of the P-glycoprotein transporter may be implicated in SE pharmacoresistance to second-line ASM in SE; however, such a conclusion cannot be made without corroborating evidence in human SE, as the function of P-glycoprotein is species-specific [221]. Earlier blockade of neuroinflammation has been found to ameliorate the severity of SE, reduce BBB damage and subsequent neurodegeneration in SE models [61,87,89]. Clinically, anti-inflammatory drugs show potential in new-onset refractory status epilepticus (NORSE), which is associated with primary neuroinflammatory processes more so than other etiologies of SE; efficacy of anti-inflammatory drugs cannot be extrapolated to all etiologies of SE [222,223]. Intrathecal dexamethasone has been shown to improve seizure control in patients with NORSE, although its effect on brain atrophy was not reported [223,224]. Similarly, anakinra, an IL-1 antagonist, has been shown to be effective in reducing seizure activity and seizure recurrence in two case reports of NORSE; however, it did not prevent brain atrophy, which may be related to late administration of the drug in both reports or additional pathological processes related to SE or NORSE [225,226]. Neuroinflammation is further exacerbated by BBB disruption, subsequent albumin extravasation, and infiltration of lymphocytes, thereby inducing further release of pro-inflammatory cytokines and progressive synaptic dysregulation [82,108]. Excitatory synaptic dysregulation triggers influx of Ca^2+^ into neurons via NMDA and AMPA receptors [52,126,132]. Ca^2+^ overload induces free radical formation and aberrant activation of second messengers and enzymes, leading to the initiation of proapoptotic pathways [38,126]. NMDA receptor-mediated Ca^2+^ influx induces oxidative stress via activating NOS and sequestration of Ca^2+^ within mitochondria [128]. NO reacts with ROS to form peroxynitrite, which mediates a myriad of aberrant processes, including apoptosis and necrosis, depending on ATP availability [38,52,126,149,150]. Ca^2+^-overload is thought to activate neurotoxic signal-transduction pathways via excitotoxic and non-excitotoxic mechanisms. It may be argued that the focus on Ca^2+^ influx through NMDA and AMPA receptors may detract from Ca^2+^-mediated non-excitotoxic mechanisms of neuronal injury. One such mechanism might involve signaling cascades downstream from neuronal P2X receptor activation, which increases neuronal permeability to Ca^2+^. P2X receptor is activated in conditions of sustained extracellular ATP or ADP levels, the latter is likely implicated in SE, though further research is needed to delineate those mechanisms [126]. ATP availability is jeopardized by supply-demand mismatch, especially in the presence of mitochondrial dysfunction [28,117]. Mitochondria are not the primary source of ROS but rather a primary target of oxidative stress, which triggers the opening of the mitochondrial permeability transition pore and the initiation of a cascade culminating in mitochondrial failure [142]. Salvaging mitochondrial function via exogenous mitochondrial transplantation and mitochondrial substrate pyruvate has been found to blunt oxidative stress, neuroinflammation, and neuronal injury [28,83]. The administration of antioxidants, such as coenzyme Q10, has had a similar effect in animal models [151]. However, the translational potential in patients is limited due to low stability, poor brain permeability, or a lack of chain-breaking antioxidant activity in hydrophilic compounds [142,145]. Neuronal death pathways are initiated secondary to excitotoxicity, mitochondrial failure, and ATP depletion [61,122,162,163]. Ferroptosis is increasingly recognized as a central pathogenic process in neurodegeneration. It is intricately associated with neuroinflammation and ROS, both as a consequence and a facilitator [163,227].The use of ferroptosis inhibitor ferrostatin-1 reduced seizure activity and seizure recurrence in chemically induced SE, highlighting the potential therapeutic value of modulating iron homeostasis in SE [228].

Loss of neurons and breakdown of neuronal networks induce synaptic and axonal remodeling, predisposing to enhanced synchronization and homogeneity of neuronal circuits, thereby rendering neuronal circuits prone to seizure [168]. Based on the similarities between the changes seen in the latent phase of SE-induced epileptogenesis and those seen in cortical undercut models and chronic GABA infusion models [43,201], we hypothesized the possibility of deafferentation-induced up-regulation of neuronal excitability. In deafferentation-induced epileptogenesis, maladaptive HSP upregulates excitatory synapses in response to prolonged network silence, thereby promoting epileptogenesis [201]. Administration of ASMs following traumatic brain injury is significantly more effective in controlling early seizures than late seizures [201]; it is possible that prolonged ASM use can trigger epileptogenesis. In light of this, further experimental research is needed to inform appropriate ASM use in post-SE so as to prevent possible HSP-mediated epileptogenesis.

It is worth reiterating that much of our understanding of the pathophysiology of SE is based on rodent models of chemical and electrical-induced seizures. Although these models have high face validity and predictive validity in that the seizures closely mirror those in human CSE and the response to therapy in those models often reflects successful translation in human SE, they lack construct validity as the mechanisms underlying induced seizures may be different from those in humans. Canine and non-human primate models have been proposed as more ideal models of naturally occurring seizures and, therefore, high construct validity [229]. However, their low cost-effectiveness and ethical concerns associated with their use limit their utilization [27].

## 7. Conclusions

Despite their limitations, rodent models of SE have contributed significantly to our understanding of the pathophysiology of SE and helped identify therapeutic targets in the search for more effective treatment. Ca^2+^ overload underlies several interlinked mechanisms of neuronal death and epileptogenesis; further research into Ca^2+^-related intracellular signaling and regulatory pathways may yield new therapeutic targets. Given the time-dependent nature of SE maladaptive alterations, the aim should be to achieve early seizure termination, whether in the hospital setting using intravenous formulations or in the prehospital setting using non-intravenous formulations. In view of the literature presented here, this review echoes the recommendations of previous research to reconsider the current guidelines of sequential therapy and examine simultaneous polytherapy in clinical trials.

## Figures and Tables

**Figure 1 ijms-26-07502-f001:**
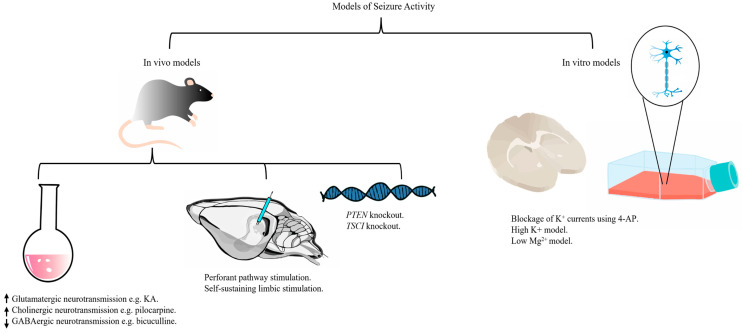
Schematic representation of the main models used to study seizure activity. In vivo models of seizures are primarily rodents, in which seizures can be divided into induced or genetic. Seizures are more commonly induced chemically by using agents that promote glutamatergic neurotransmission, either directly, such as KA, or indirectly, such as pilocarpine, or electrically by electrode stimulation of the hippocampus. In vitro preparations are predominantly of brain slices obtained from rodent models in which epileptiform activity can be induced by ionic or pharmacological means.

**Figure 2 ijms-26-07502-f002:**
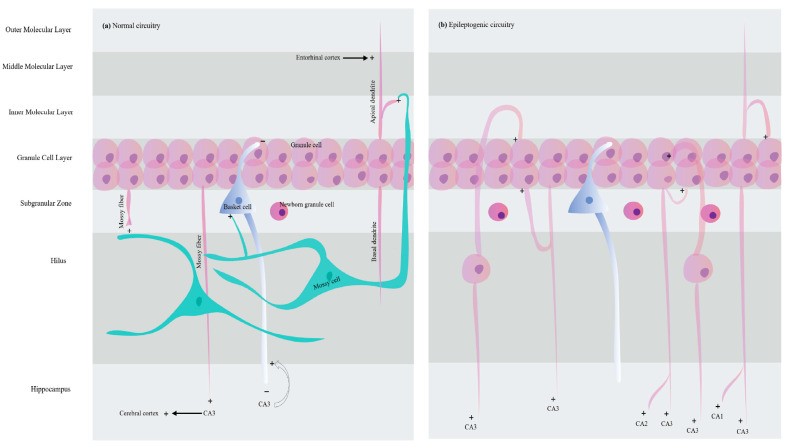
Schematic of hippocampal dentate gyrus circuitry before and after epileptogenesis: (**a**) Granule cells project mossy fibers to pyramidal neurons in area CA3, interneurons in the dentate hilus, as well as mossy cells. Mossy cells excite basket cells, which then provide feedback inhibition to granule cells; (**b**) During epileptogenesis, deafferentation of basket cells in the dentate hilus due to mossy cell death leads to disinhibition of granule cells and is associated with increased neurogenesis and increased mossy fiber sprouting from granule cells and ectopic granule cells resulting in increased connectivity to CA3 pyramidal cells and recurrent excitatory collaterals to granule cells forming monosynaptic interconnections to granule cells in place of the synaptic pathways formerly originating from mossy cells. Mossy fiber sprouting also forms connections to pyramidal cells in CA2 and CA1; (+) and (−) indicate excitatory (glutamatergic) and inhibitory (GABAergic) synapses, respectively.

**Figure 3 ijms-26-07502-f003:**
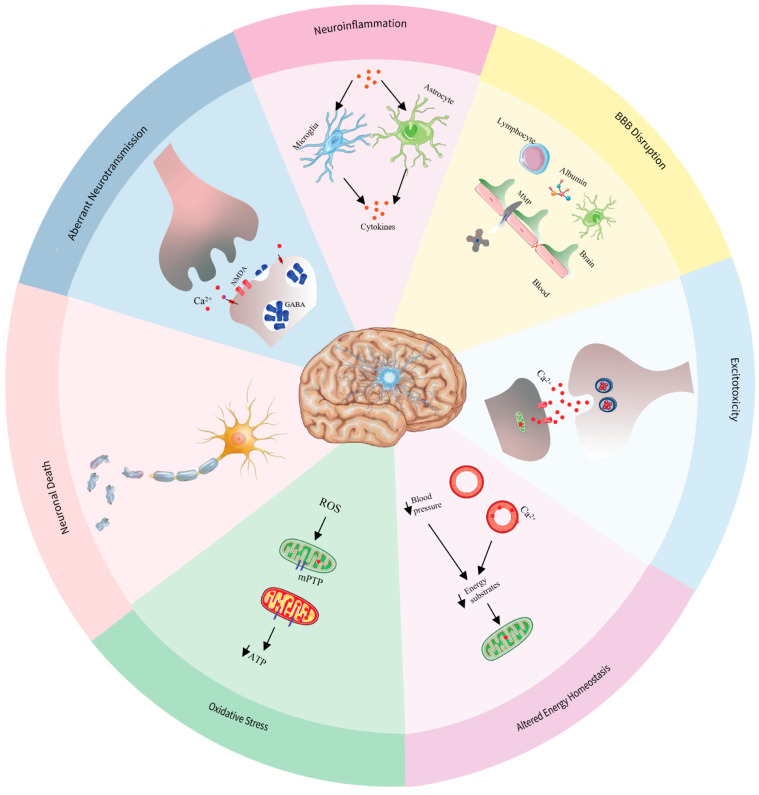
Schematic representation of the main pathophysiological mechanisms behind status epilepticus. Excessive excitation and activation of glutamate receptors cause excessive influx of Ca^2+^ and efflux of K^+^ followed by internalization of GABAergic receptors and synaptic presentation of glutamatergic receptors, leading to a cycle of synaptic potentiation which is propagated by the release of inflammatory mediators that induce astrogliosis which then stimulates the release of those inflammatory mediators and the release of ET1 which increases BBB permeability and the activation of MMPs. BBB disruption permits the infiltration of immune cells into the brain and propagation of the inflammatory cycle, as well as excitotoxicity through albumin binding to astrocytic receptors. Neurovascular and neurometabolic decoupling associated with seizure prolongation compromises the availability of energy substrates required for homeostatic pump activity and physiologic neurotransmission, exacerbating excitotoxicity and excessive Ca^2+^ influx, which disrupts mitochondrial function and ATP production. This is associated with increased ROS production, which mediates several cytotoxic sequelae, including potentiation of mitochondrial permeability transition pore opening, thereby potentiating Ca^2+^-mediated mitochondrial membrane depolarization leading to further ATP depletion. The aforementioned mechanisms culminate in activated neuronal death pathways and subsequent death of affected neurons by necrosis and apoptosis.

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
