# Peer review of "Pathophysiology of Status Epilepticus Revisited"

_ijms, 2025, doi:10.3390/ijms26157502_

Round 1

Reviewer 1 Report

Comments and Suggestions for Authors

I consider this work to be an excelent and exhaustive review of the various biological and molecular mechanisms involved in the development of status epilepticus and related to drug-resistant epilepsies. However, it requires reviewing some concepts and including mechanisms not mentioned in the text, as indicated as follow:

ijms-3748012. Pathophysiology of Status Epilepticus Revisited

1-Definition of SE: Text in Abstract:

“Status epilepticus is a hyperexcitable, hypersynchronous neuronal discharge resulting 9 from an imbalance between excitatory and inhibitory mechanisms”.

Comment 1: This phrase is the definition of a seizure, but not of status epilepticus

and should be changed to: “Status epilepticus occurs when a seizure lasts more than 5 minutes or when several seizures occur very close between them and the patient doesn't recover consciousness during interictal times”

2- Text in abstract

“Oxidative stress resulting from mitochondrial leak and increased production of reactive oxygen species activates the inflammasome and induces a damage-associated molecular pattern”, which can lead to cell death by ferroptosis

Comment 2: add the text in red

See references:

Fikry H, Saleh LA, Mahmoud FA, Gawad SA, Abd-Alkhalek HA. CoQ10 targeted hippocampal ferroptosis in a status epilepticus rat model. Cell Tissue Res. 2024;396(3):371-397. doi: 10.1007/s00441-024-03880-z

Du K, He M, Zhao D, Wang Y, Ma C, Liang H, Wang W, Min D, Xue L, Guo F. Mechanism of cell death pathways in status epilepticus and related therapeutic agents. Biomed Pharmacother. 2022;149:112875. doi: 10.1016/j.biopha.2022.112875.

3-Text Introduction, lines 35-36.

“When a seizure is no longer self-terminating or is repetitive without a complete return to baseline, it is called status epilepticus (SE)”

  • Comment 3: This text must be changed according to the definition indicated in comment 1 and the League Against Epilepsy (ILAE) task force that define SE as:
  • 5 minutes for generalized tonic-clonic seizures
  • 10 minutes for focal seizures
  • 10 to 15 minutes for absence seizures

(see: https://www.ilae.org/journals/epigraph/epigraph-vol-20-issue-2-fall-2018/time-is-brain-treating-status-epilepticus)

4-line 82:

[18, Novy, J].

Comment 4: It should be changed by [18], and “Novy, J” should be eliminated.

5- AEDs

Comment 5: It should be changed to anti-seizure medication (ASM) in all the text of the manuscript.

4.1. Aberrant Neurotransmission pag 4

See reference: Fattorusso A, Matricardi S, Mencaroni E, Dell'Isola GB, Di Cara G, Striano P, Verrotti A. The Pharmacoresistant Epilepsy: An Overview on Existent and New Emerging Therapies. Front Neurol. 2021 Jun 22;12:674483. doi: 10.3389/fneur.2021.674483.

6-Page 5, lines 188-191

“Alternatively, it is suggested that increased [K+]o from excessive neuronal release during discharges and reduced buffering by astrocytes, due to reduction of by astrocytic inwardly rectifying potassium (Kir) channels containing Kir4.1 subunits (Kir4.1 channels), can alter the resting membrane potential and block neuronal  depolarization, leading to seizure termination”……….

Lines199-206 “In these cases, surround inhibition may be overcome by enhanced neurotransmitter release due to excessive presynaptic accumulation of Ca2+ in presynaptic terminals and excessive elevation of [K+]o, which, rather than block depolarization, would dampen the inhibitory, hyperpolarizing outward K+ currents. Although an increased level of [K+]o normally prevents action potential propagation, misbalanced polarity, wherein the membrane potential of a neuronal population is more hyperpolarized compared to surrounding areas, creates conditions for the generation of seizures”

Comment 6: These paragraphs are extremely important but require a clearer explanation of Kir channels role.

See the following concepts

Loss of Kir4.1 causes membrane depolarization, more seizures and a break-down of K+ and glutamate homeostasis which results in seizures and wide-spread white matter pathology, and additionally similar loss expression of other Kir channels in heart was related with SUDEP in an experimental model’s seizures.

See references

a-Frigerio F, Frasca A, Weissberg I, Parrella S, Friedman A, Vezzani A, Noé FM. Long-lasting pro-ictogenic effects induced in vivo by rat brain exposure to serum albumin in the absence of concomitant pathology. Epilepsia. 2012 Nov;53(11):1887-97. doi: 10.1111/j.1528-1167.2012.03666.x

b-Olsen ML, Sontheimer H. Functional implications for Kir4.1 channels in glial biology: from K+ buffering to cell differentiation. J Neurochem. 2008 Nov;107(3):589-601. doi: 10.1111/j.1471-4159.2008.05615.x

c-Auzmendi J, Akyuz E, Lazarowski A. The role of P-glycoprotein (P-gp) and inwardly rectifying potassium (Kir) channels in sudden unexpected death in epilepsy (SUDEP). Epilepsy Behav. 2021 Aug;121(Pt B):106590. doi: 10.1016/j.yebeh.2019.106590.

7-Text page 5, line 199: “Additionally, low serum concentration of AEDs in patients with epilepsy as an etiology of SE is associated with better outcome than other etiologies [17].

Comment 7: The article indicated in reference 17, by Chenand Wasterlain, mentions the preference for drugs with a short elimination half-life and discusses some therapeutic choices, but the definition of a better prognosis in cases of subtherapeutic ASM levels as a cause of SE, compared to other causes of SE, is not clear.

In this regard, the next consideration by Barry and Hauser, could be better: “Citing ASM irregularity as the major cause of SE in patients with epilepsy oversimplifies a complex, poorly understood situation” (see reference Barry E, Hauser WA. Status epilepticus and antiepileptic medication levels. Neurology. 1994 Jan;44(1):47-50. doi: 10.1212/wnl.44.1.47)

8-Page 7, lines 299-301.

Text: “M1 microglia exert a similar neurotoxic effect through promoting the transcription of M1-associated genes, including inducible nitric oxide synthase (iNOS), cyclooxygenase-2 (COX-2), and HMGB1 [71].”…..

“Another molecule heavily implicated in the propaga-333 tion of SE is the prostaglandin COX2, which is released by hyperexcitable neurons and 334 M1 microglia [4,87]. Upregulation of COX-2 in hippocampal neurons is evident in the 335 early hours of SE onset, far preceding neuronal death in in vivo and in vitro SE models, 336 suggesting that COX-2-mediated inflammation is a reaction to seizure activity rather than 337 neuronal death [88,89].

Comment 8: The neurotoxic role of microglia also corresponds to a mechanism known as ferroptosis, which is not mentioned by the authors, and which has already been described to be induced by epileptic status, both at the cerebral and cardiac levels, with a potential relationship with SUDEP.

At the same time, two inflammatory mediators are mentioned, such as COX 2 and HMGB1, which induce the expression of ABC-transporter P-glycoprotein (see reference 87), one of the main players in drug resistance in epilepsy. This topic is not considered or mentioned in the manuscript and should be included. It has already been shown that the greater the load of convulsive stress, the higher the level of expression of this transporter, not only at the cerebral level, which is related to an increase in the severity of seizures (including epilepsy), but also at the cardiac level.

See refences:

  • Liddell JR, Hilton JBW, Kysenius K, Billings JL, Nikseresht S, McInnes LE, Hare DJ, Paul B, Mercer SW, Belaidi AA, Ayton S, Roberts BR, Beckman JS, McLean CA, White AR, Donnelly PS, Bush AI, Crouch PJ. Microglial ferroptotic stress causes non-cell autonomous neuronal death. Mol Neurodegener. 2024 Feb 5;19(1):14. doi: 10.1186/s13024-023-00691-8.
  • Moscovicz F, Taborda C, Fernández F, Borda N, Auzmendi J, Lazarowski A. Ironing out the Links: Ferroptosis in epilepsy and SUDEP. Epilepsy Behav. 2024 Aug;157:109890. doi: 10.1016/j.yebeh.2024.109890.
  • Akyüz E, Saleem QH, Sari Ç, Auzmendi J, Lazarowski A. Enlightening the mechanism of ferroptosis in epileptic heart. Curr Med Chem. 2023 Feb 23. doi: 10.2174/0929867330666230223103524
  • Czornyj L, Auzmendi J, Lazarowski A. Transporter hypothesis in pharmacoresistant epilepsies. Is it at the central or peripheral level? Epilepsia Open. 2022 Aug;7 Suppl 1(Suppl 1):S34-S46. doi: 10.1002/epi4.12537. Epub 2021 Oct 29. PMID: 34542938
  • Akyuz E, Doganyigit Z, Eroglu E, Moscovicz F, Merelli A, Lazarowski A, Auzmendi J. Myocardial Iron Overload in an Experimental Model of Sudden Unexpected Death in Epilepsy. Front Neurol. 2021 Feb 11;12:609236. doi: 10.3389/fneur.2021.609236
  • Xie Y, Yu N, Chen Y, Zhang K, Ma HY, Di Q. HMGB1 regulates P-glycoprotein expression in status epilepticus rat brains via the RAGE/NF-κB signaling pathway. Mol Med Rep. 2017 Aug;16(2):1691-1700. doi: 10.3892/mmr.2017.6772
  • van Vliet EA, Zibell G, Pekcec A, Schlichtiger J, Edelbroek PM, Holtman L, Aronica E, Gorter JA, Potschka H. COX-2 inhibition controls P-glycoprotein expression and promotes brain delivery of phenytoin in chronic epileptic rats. Neuropharmacology 2010;58(2):404-12. doi: 10.1016/j.neuropharm.2009.09.012
  • Bauer B, Hartz AM, Pekcec A, Toellner K, Miller DS, Potschka H. Seizure-induced up-regulation of P-glycoprotein at the blood-brain barrier through glutamate and cyclooxygenase-2 signaling. Mol Pharmacol. 2008 May;73(5):1444-53.

doi: 10.1124/mol.107.041210.

  • Robey RW, Lazarowski A, Bates SE. P-glycoprotein--a clinical target in drug-refractory epilepsy? Mol Pharmacol. 2008 May;73(5):1343-6.

doi: 10.1124/mol.108.046680

Author Response

Comment 1: edited.

 Comment 2: We’ve mentioned that oxidative stress leads to necrosis and apoptosis as well. Therefore, we couldn’t focus solely on ferroptosis in the abstract, but we did incorporate some relevant information on ferroptosis in the text, specifically at lines 295, 384, 566, and 893.

 Comment 3: We edited the definition and added the ILAE definition under the definition section.

Comment 4: noted

 Comment 5: noted 

Comment 6: We appreciate your providing the references, which have been very helpful to correct the concept that you’ll find in lines 192 and 202.   

Comment 7: noted 

Comment 8: The references have been very helpful in helping us integrate p-gp’s role in lines 357 and 842.  

Reviewer 2 Report

Comments and Suggestions for Authors

This article by Alsherhri et al reviews on the pathoohysiological mechanism of status epilepticus (SE) underlying its self-sustainability, pharmacoresistance and subsequent epileptogeneis, with emphasis on the findings of previous studies on animal models of SE.

In general, this paper is well written, informative and interesting. Prior to publication, several minor issues should be addressed:

  1. In many of the previous studies, the reported case fatality rate of SE was not so high as “between 20 and 30%” (lines 39-40).
  2. The duration of established SE is described as “more than 30 minutes” in Section 2 (lines 78-80), but as “within 10-30 minutes” in Section 4 (lines 233-235). Which is correct?
  3. With regard to the treatment of NORSE with anti-inflammatory drugs, some recent papers on the use of anakinra and intrathecal dexamethasone could be cited.
  4. In the latter half of Discussion (lines 814-880), the reviewer would see more description on the currently ongoing therapeutic trials and future prospects of therapies.
  5. The comment “pharmacological research should focus on developing non-intravenous formulations of non-BZD AEDs” looks unexpected and requires further explanation.
  6. The use of abbreviations that are either unfamiliar to many readers or only infrequently used, such as SRS (spontaneous recurrent seizures), UPS (ubiquitin-proteasome system) and mPTP, could be avoided.
  7. Some of the cited papers, such as Novy J (line 82) and Miyake, T (line 296), do not appear in the list of References.
  8. Syntax and typographic errors: (hyperexcitability) and (hypersynchrony) (lines 162-164; unnecessary parentheses?), reduction of by astrocytic inwardly rectifying potassium channels (lines 188-189), O2 (line 403; superscript→subscript), To which the brain is particularly sensitive (line 510), different among studies, (line 607, comma -> semicolon?), Na+ (line 746, →superscript), which triggers the opening of mPTP and a cascade trigger (line 833).

Author Response

  1. We corrected the references (2 and 5).
  2. 30 minutes is the official time point. We edited section 4.
  3. True. The discussion now includes a mention of anakinra and dexamethasone, in line 859.
  4. Noted. We have added some relevant updates to the discussion, which we hope will be sufficient.
  5. True. We edited this and included it in the discussion.
  6. Edited.
  7. Edited.
  8. Edited.

Round 2

Reviewer 1 Report

Comments and Suggestions for Authors

The modifications have been satisfactory and in accordance with the suggestions made- There are some typos in the bibliographic citations. For example,

ref- 52. 3Wasterlain,

ref. 102 is misaligned,

ref. 104. 3Marchi, N.